# Crop Disease Identification by Fusing Multiscale Convolution and Vision Transformer

**DOI:** 10.3390/s23136015

**Published:** 2023-06-29

**Authors:** Dingju Zhu, Jianbin Tan, Chao Wu, KaiLeung Yung, Andrew W. H. Ip

**Affiliations:** 1School of Computer Science, South China Normal University, Guangzhou 510631, China; 2021023224@m.scnu.edu.cn; 2School of Software, South China Normal University, Guangzhou 510631, China; 2022024225@m.scnu.edu.cn; 3Department of Industrial and Systems Engineering, Hong Kong Polytechnic University, Hong Kong 999077, China; kl.yung@polyu.edu.hk; 4Department of Mechanical Engineering, University of Saskatchewan, Saskatoon, SK M4Y1M7, Canada; wh.ip@polyu.edu.hk

**Keywords:** convolutional neural network, vision transformer, self-attention mechanism, image classification, crop disease recognition

## Abstract

With the development of smart agriculture, deep learning is playing an increasingly important role in crop disease recognition. The existing crop disease recognition models are mainly based on convolutional neural networks (CNN). Although traditional CNN models have excellent performance in modeling local relationships, it is difficult to extract global features. This study combines the advantages of CNN in extracting local disease information and vision transformer in obtaining global receptive fields to design a hybrid model called MSCVT. The model incorporates the multiscale self-attention module, which combines multiscale convolution and self-attention mechanisms and enables the fusion of local and global features at both the shallow and deep levels of the model. In addition, the model uses the inverted residual block to replace normal convolution to maintain a low number of parameters. To verify the validity and adaptability of MSCVT in the crop disease dataset, experiments were conducted in the PlantVillage dataset and the Apple Leaf Pathology dataset, and obtained results with recognition accuracies of 99.86% and 97.50%, respectively. In comparison with other CNN models, the proposed model achieved advanced performance in both cases. The experimental results show that MSCVT can obtain high recognition accuracy in crop disease recognition and shows excellent adaptability in multidisease recognition and small-scale disease recognition.

## 1. Introduction

Diseases have always been an important disadvantage in agricultural production. Early detection and removal of diseases are effective means to improve the quality of crop growth. In traditional agricultural production, the identification of crop diseases is mainly performed through manual analysis combined with professional knowledge. The manual identification of diseases is inefficient, is error-prone, requires high expertise, and is costly in terms of human resources [1]. Therefore, it is the goal of many agricultural researchers to minimize human intervention and to achieve an intelligent diagnosis of diseases. In the early days of agricultural production, many researchers used machine learning to solve problems in the field. Multivariate adaptive regression splines (MARS) is a nonparametric machine learning method, which has the advantage of being able to deal with large amounts of data and high-dimensional data, fast calculation, and an accurate model. Akin and Eyduran et al. used MARS to incorporate nominal variables as predictor variables into the inherent properties of the model to determine the major nutritional requirements of optimal major salts for good branch quality, reproduction, and greener foliage of strawberry varieties [2]. Random forest is an ensemble learning method based on bagging, which can deal with classification and regression problems and high-dimensional data well, and is not easy to overfit. Jun and Fang et al. extracted features in the segmented disease area as the input of the random forest algorithm, and then made the comprehensive accuracy of four common sunflower diseases’ recognition reach 95.0%. There is also the KNN (k-nearest neighbor) algorithm, which can be used for classification and regression problems on small datasets [3]. Gaikwad et al. reviewed various classification techniques used to identify leaf diseases and concluded that KNN is the simplest in predicting a class of test examples and is easy to implement and understand [4]. In general, although the method of machine learning has good performance in agriculture, there are still shortcomings; for example, the accuracy and rate of classification, detection, and recognition are not high enough, and there is no good robustness and versatility. For example, the quality of the MARS model depends on whether the preset maximum number of basis functions is reasonable, the random forest will be affected in low-dimensional small datasets and datasets with different values of attributes, and the high time complexity of prediction and sensitivity to noise and outliers are the shortcomings of KNN.

In recent years, with the development of deep learning [5], deep learning models based on a convolutional neural network (CNN) [6] have achieved excellent performance in the field of image recognition [7]. In the 2012 ILSVRC competition, AlexNet, designed by Hinton and his student Alex Krizhevsky, achieved the best recognition accuracy on the ImageNet dataset [8]. In 2014, GoogleNet, proposed by Google’s team, also achieved the best result in the ImageNet competition [9]. Since CNN has shown excellent performance in image recognition, it has been widely used in various fields, such as medical image recognition [10], face recognition [11], animal classification [12], and vegetation monitoring [13].

Some researchers have also applied CNN models to agricultural disease recognition with good results. Mohanty et al. adapted the pretrained models AlexNet and GoogleNet, using transfer learning, based on the PlantVillage dataset containing 14 crops and 26 diseases. Both models achieved more than 99% accuracy on the testing set, demonstrating the effectiveness of deep learning models in intelligently diagnosing crop diseases [14]. Zhang et al. achieved 98.9% accuracy of the improved GoogLeNet by optimizing the parameters of GoogLeNet, adding dropout, and adjusting the number of classifiers for the identification of eight maize leaf diseases [15]. Suryawati et al. used CNN models with different structures for tomato disease identification. The experimental results showed that the VGG model with a deeper structure achieved the best testing accuracy [16]. Mukti and Biswas compared the pretrained models ResNet50, VGG16, VGG19, and AlexNet on a large plant leaf dataset containing 70,295 training images and 175,772 validation images. The results showed that the fine-tuned ResNet50 model achieved the best performance of 99.80% accuracy [17]. Qiu et al. applied the pretrained network VGG16 to rice disease identification [18] based on a dataset containing 10 rice diseases and improved the model using parameter fine-tuning and a linear discriminant classifier. Both the training accuracy and prediction accuracy reached over 96%, providing a good prediction model for rice disease identification [19]. These researchers have improved on the traditional CNN model and achieved good results on a variety of crop datasets.

To solve the problem that heavyweight models are difficult to be applied to agricultural production, some researchers have improved some existing lightweight models or designed their model structures with simple structures and a small number of parameters according to the characteristics of crop disease datasets and obtained good recognition performance. Kamal et al. proposed a lightweight model, Reduced MobileNet, with a deep separable convolutional structure for leaf disease recognition [20]. They trained the model on a large dataset of 82,161 leaves containing 55 different classes of plants. Experimental results show that Reduced MobileNet achieves 98.34% classification accuracy, achieving disease recognition ability comparable to the original MobileNet. As a result, Reduced MobileNet can meet the real-time crop recognition of mobile devices with its smaller model size while maintaining higher accuracy. In another study on optimizers for MobileNetV2, Zaki et al. extracted 4671 tomato disease images from the PlantVillage dataset as an experimental dataset and trained the MobileNetV2 model using five optimizers [21]. The results showed that MobileNetV2 obtained the best classification performance when the optimizer Adagrad was used. Additionally, in a recent study, Sutaji and Yıldız proposed LEMOXINET, a high-precision crop disease identification model fusing MobileNetV2 and Xception, which effectively merges the leaf features extracted by MobileNetV2 and Xception [22]. It is shown that LEMOXINET achieves 99.10% accuracy on the PlantVillage dataset and 99.52% accuracy on a self-collected dataset containing multiple crop diseases. In 2022, Shifeng Dong et al. proposed a YOLOv4-based crop pest detection framework, YOLO-pest, which uses MobilenetV3 instead of the YOLOv4 backbone network to significantly reduce the number of parameters, in response to the significant decrease in pest detection accuracy of YOLO when dealing with pest datasets with large-scale variations and multiple classes [23]. In the same year, Wei Zhang et al. proposed a lightweight pest detection model, AgriPest-YOLO, for achieving a good balance between efficiency, accuracy, and model size for pest detection. The model proposes a coordination and local attention (CLA) mechanism to obtain richer and smoother pest features with less interference from noise [24]. Their proposed approach further improves the identification of lightweight models on crop disease datasets.

CNN-based deep learning models have achieved excellent results in a crop disease identification work, thanks in large part to their ability to extract local information efficiently. In many cases, however, obtaining larger sensory fields can often only be achieved by using large convolutional kernels or stacking multiple small convolutional kernels in succession [25]. This characteristic determines that CNN has an inherent disadvantage in modeling global information. CNN, while effectively learning local features, is unable to model the relationship between the different regions in an image. In the problem of global information extraction, the vision transformer (ViT) [26] proposed in recent years provides a feasible solution. Transformer [27] was originally proposed in the field of NLP, and its core module is the self-attention mechanism. The proposed mechanism can capture the long-range relative relationships of word vectors in text information processing and thus resolve global features of the whole text data. Inspired by Transformer, some researchers tried to apply Transformer to the field of vision and proposed ViT. However, it does not have the translational invariance of CNN, which makes it require a huge amount of data to have a performance beyond that of CNN. More recently, some researchers have proposed combining the benefits of CNN and Transformer to propose a few hybrid ViT models, which capture long-range dependencies by embedding a layer of self-attention in the CNN structure. For example, BoTNet [28] adds multihead self-attention to the last three bottleneck blocks of ResNet to form a new network structure, Bottleneck Transformer. The experimental results show that BoTNet achieves a top 1 accuracy of 84.7% on the ImageNet-1K dataset, which is higher than the ViT model using the Transformer structure alone. Another study by MobileViT [29] used MobileNetV2’s inverted residual block as part of its structure for local feature extraction of images and used Transformer to replace local modeling in the convolution with global modeling. On the ImageNet-1K dataset, MobileViT achieves a top 1 accuracy of 78.4% with about 6 million parameters, which is 3.2% and 1.0% more accurate than ResNet101 and MobileNetV3.

All of these models achieved performance beyond that of traditional CNN models on publicly available large image datasets. However, there is still a lack of relevant studies combining a self-attention mechanism to design models with high recognition accuracy on smaller-scale agricultural disease datasets. The self-attention mechanism brings new possibilities for a deep learning model design in agriculture. Based on the current research status of CNN models in the field of agricultural disease identification and the advantageous performance of ViT models in constructing long-range feature relationships, we summarize the limitations of previous studies and the main contributions of our work from the following points:Previous disease identification models are still dominated by CNNs, which have limitations in the expansion of the receptive field and the perception of global information. Our work integrates the features of CNN and Transformer and designs a multiscale self-attention (MSSA) module for the information fusion of local and global features of disease images.For crop disease identification, the self-attention mechanism lacks practical applications to verify its effectiveness. It is still difficult to design lightweight and high-performance hybrid ViT models with real-world application datasets. We propose a hybrid model fusing multiscale convolution and vision transformer (MSCVT) for crop disease identification based on the ResNet five-stage architecture and insert MSSA modules in each stage. In addition, we validated the effectiveness of the proposed model on two disease datasets with different scales.To ensure that the proposed model is lightweight, we introduce the inverted residual block to replace the ordinary convolution of 3×3. In the end, MSCVT has only 4.20 M parameters and achieved 99.86% and 97.50% recognition accuracies on the PlantVillage test dataset and the Apple Leaf Pathology test dataset, respectively.

## 2. Material and Methods

### 2.1. Model Design

As shown in Figure 1, our model is designed with reference to the five-stage architecture of ResNet. The original images are uniformly adjusted to 224×224×3 and input to the model. In stage 1, the images are first passed through 7×7 convolution and a maxpooling layer for two consecutive downsampling operations. The reason is that the combination of 7×7 convolution and the maxpooling layer can reduce the feature map size while maintaining a large receptive field. In addition, the reduced size of the feature map can keep the number of parameters small in the later stage. The downsampled feature maps go through four stages for local and global information extraction and fusion. A stage consists of a downsampling layer and an MSSA module. The extraction of local information is mainly performed by the inverted residual block and the multiscale convolution layer in the MSSA module. The downsampling layer is also composed of an inverted residual block with the stride of 2, which completes the feature map size reduction and channel expansion in each stage. In order to avoid losing a large amount of shallow information due to multiple downsampling, no downsampling is performed in stage 2, and the stride of the corresponding 3×3 convolution is set to 1. In the process of feature extraction from shallow to deep layers, each stage obtains a different resolution of the feature map due to the presence of the downsampling layer. Therefore, the self-attention mechanism in ViT is added to the MSSA module in each stage to learn the information of different global receptive fields.

Our model uses the average pooling layer instead of the fully connected layer. In earlier networks, such as AlexNet, VGG, etc., the fully connected layer takes up nearly 80% of the total number of parameters in the network, which makes the model itself very bulky. An average pooling layer accomplishes the role of dimensionality reduction by averaging each feature map, and also significantly reduces the number of parameters [30]. Finally, the Fc layer is used to map the output of the previous layer into a one-dimensional vector of length of the number of classifications. The parameters of each stage of the model are shown in Table 1.

### 2.2. Inverted Residual Block

The inverted residual (IR) block is a lightweight network structure proposed by MobileNetV2, which is based on MobileNetV1’s deep separable convolution (DSC) improvement.The advantage of DSC is the ability to obtain good information aggregation while maintaining a low number of parameters and computational effort [31]. DSC divides an ordinary convolution process into two steps, depthwise convolution (DC) and pointwise convolution (PC). One convolution kernel of DC is responsible for one channel of the input feature map, and each channel of the feature map is computed independently of the other in this process. Therefore, DC only computes in the spatial dimension, and the number of output feature maps is the same as the number of channels in the input layer, and there is no information interaction between channels, while PC accomplishes the task of combining the DC-generated feature maps in the depth direction to complete the feature fusion in the channel dimension. Assuming that the number of input channels is A and the number of input channels is B, the number of parameters of S × S DSC is the sum of the number of parameters of DC and the number of parameters of PC, which can be calculated as Equation (Equation 1):(1)$${\mathrm{Param}}_{\mathrm{DSC}}=A*S*S+A*1*1*B$$

The number of parameters of the ordinary convolution is A × S × S × B. It can be obtained that DSC converts some of the multiplication operations in ordinary convolution into addition operations, thus greatly reducing the computational complexity. In addition, the outputs of both DC and PC are input to the batch normalization and ReLu activation functions to improve the convergence speed during training and enhance the nonlinear representation of the model. The structure of DSC is shown in Figure 2.

The residual block has been proven in ResNet to help improve the accuracy to build deeper networks, so MobileNetV2 also introduces a similar block. The classical process of the residual block is to downscale by 1×1 convolution, then to extract features by 3×3 convolution, and finally to upscale by 1×1. However, the depthwise convolution layer extracts features limited to the input feature dimension. If the residual block is used, the input feature map is first compressed by the 1×1 pointwise convolution operation, and then the features extracted by the depthwise convolution will be less. Therefore, MobileNetV2 first expands the channels of the feature map by 1×1 pointwise convolution operation to enrich the number of features and improve the accuracy. This process just reverses the order of the residual block; i.e., it first ascends through 1×1 convolution, then performs feature extraction through 3×3 convolution, and finally descends through 1×1. According to the parameter configuration of MobileNetV2, the multiplier for channel expansion is set to 6. The inverted residual block is shown in the Figure 3.

### 2.3. Self-Attention Module

The self-attention mechanism first appeared in Transformer, proposed by Ashish Vaswani et al. It is based on the idea of obtaining long-distance feature relationships by computing the similarity between words in a sentence, and has been widely used in text classification [32] and sentiment analysis [33]. Unlike serialized vectors, the information of an image is stored in the form of a matrix [34]. This means that it is not possible to apply Transformer directly to the field of vision. In order to allow images to also acquire long-distance information on images like serial vectors, Alexey Dosovitskiy et al. proposed ViT. Their approach is to slice the image into fixed-size patches, and then obtain the patch embedding by a linear transformation. As shown in Figure 4, to divide the image into P × P patches, the image is first cut into $\frac{\mathrm{HW}}{P^{2}}$ patches by reshaping. Then each patch is stretched into a one-dimensional vector of the length $P^{2}\;\times$ C. Finally, the patches are converted into patch embedding of the specified length by a linear transformation, which is similar to word embedding in NLP. Each patch embedding is equivalent to a word embedding in a sentence, and the whole image is equivalent to a sentence. After obtaining the patch embedding, we also need to embed the position information of each patch by position embedding. In NLP, position embedding is used to add position information to words [35]. Similarly, it is possible to embed position information to the patch embedding in ViT. A straightforward way is to generate a learnable parameter with the same dimension as the patch embedding as the positional embedding randomly, and then add the patch embedding and the positional embedding directly. The input of the final self-attention mechanism calculation is shown in Equation (Equation 2).
(2)$${\mathrm{Input}}_{\mathrm{Patch}}=\mathrm{Patch}\;\mathrm{Embedding}\left(x\right)+\mathrm{Position}\;\mathrm{Embedding}$$

As shown in Figure 5, the input image is first converted into patch embedding and embedded in position embedding, and then the self-attention mechanism can be computed. Each Input_Patch will be multiplied with three transformation matrices, $W^{Q}$, $W^{K}$, and $W^{V}$, to obtain three vectors, Query (Q), Key (K), and Value (V) [36]. Then, the similarity between each patch and other patches in the image is calculated by dot product operation on Q and K. This similarity value is calculated by the softmax function to obtain a set of weights. The attention feature matrix is obtained by summing the product of these weights and the corresponding V, which represents the image information represented by each patch and the dependencies between the image regions represented by different patches. The core of the whole calculation process can be expressed as Equation (Equation 3):(3)$$\mathrm{Attention}\left(Q,K,V\right)=(\mathrm{softmax}\left(\frac{{\mathrm{QK}}^{T}}{\sqrt{d_{k}}}\right))V$$

In order to obtain complete and detailed global information, the size of the sliced patch is set to 1×1, and the length of each Input_Patch is equal to the number of channels. This means that each pixel of the image will calculate the dependency factor with the pixels in other regions, and the computational complexity of the SA module is proportional to the square of the image resolution. Therefore, we refer to the ResNet structure and perform two downsampling operations in the first stage and add downsampling layers in the following stages. After the image size is reduced several times, the number of parameters generated by the SA module is largely reduced.

### 2.4. MSSA Module

An MSSA module consists of a multiscale feature fusion layer and self-attention module (SA module), and the overall architecture is shown in Figure 6. After the feature map with the dimension H × W × C (where H, W, and C denote height, width, and the number of channels, respectively) enters the MSSA module, a larger receptive field and more detailed information are first obtained from the spatial dimension by a 3-layer inverted residual block. Then the feature map is sliced into 3 subfeature maps from the channel dimension, and the dimensions of the subfeature maps are all H × W $\times \;(\frac{1}{3C})$. We input these subfeature maps into 3 × 3, 5 × 5, and 7 × 7 ordinary convolution layers to obtain subfeature maps with different receptive fields. Finally, these subfeature maps are fused by concatenation to obtain a feature map with multiscale information in the channel dimension. The local features obtained from spatial and channelwise acquisition of the image complement each other, which can effectively enhance the disease information and suppress irrelevant information [37]. The above process can be expressed as Equation (Equation 4):(4)$${\mathrm{Output}}_{\mathrm{multiscale}}\left(z_{1},z_{2},z_{3}\right)=\mathrm{Concatenate}\left({\mathrm{Conv}}_{3\times 3}\left(z_{1}\right),{\mathrm{Conv}}_{5\times 5}\left(z_{2}\right),{\mathrm{Conv}}_{7\times 7}\left(z_{3}\right)\right)$$
where Output${}_{\mathrm{multiscale}}$ denotes the output from which multiscale information is obtained. z${}_{1}$, z${}_{2}$, and z${}_{3}$ denote the subfeature maps generated by splitting the initial input along the channel direction after 2-layer 3×3 convolution, respectively.

The self-attention mechanism is introduced to work on the extraction of global information. The traditional CNN model obtains a wider receptive field by stacking convolution blocks, and the obtained receptive field is also limited. In addition, in previous crop disease recognition studies, most of them revolve around how to obtain more detailed local information and often ignore the correlation between different locations of the images. Therefore, we input the feature map output from the multiscale convolution layers to the SA module to complement the global receptive field of the model. Due to the ability of the self-attention mechanism to capture long-range dependency, local feature blocks at different locations are correlated. Even the edge information of the image will be identified.

As the network deepens gradually, the shape of the feature map becomes progressively smaller due to multiple downsampling. Additionally, the input feature map of each stage is the result of mapping the image several times, and inputting feature maps with different resolutions in different stages will cause the region information represented by the generated patches to change. Therefore, the position information parameter position embedding and the dependencies between patches need to be relearned in each stage.

In order to make the model more expressive, we repeatedly stack the MSSA modules at each stage to form a double-loop structure. The feature map output from the MSSA module is re-entered into the MSSA module, and it is worth noting that the positional embedding in the SA module is removed when re-entered. This is because the location information of patch is already embedded in the computed result of the first MSSA module, and the calculation process of the second MSSA module is able to receive the long-range dependence information obtained from the previous calculation. This improves the reuse rate of information and reduces the number of unnecessary parameters and computations. In addition, residual connections are added to the outermost part of the entire MSSA module to prevent the occurrence of model overfitting.

### 2.5. Experimental Datasets

Experimental data were obtained from PlantVillage, a public dataset produced by the Pennsylvania State University, and the Apple Leaf Pathology dataset produced by Northwestern Agricultural and Forestry University. The PlantVillage dataset contains 12 healthy leaves and 26 diseased leaves with a total of 54,306 images. It is the most widely used public dataset for the identification of crop diseases. Table 2 shows the number of images for each crop in the PlantVillage dataset, the number of disease categories, and the number of health categories included in that crop. The Apple Leaf Pathology dataset contains 5 different apple leaf diseases, which were mostly acquired under good lighting conditions on sunny days, with a few images being acquired on rainy days. This dataset was initially composed of 1591 images, which was expanded to 26,376 images after various data augmentation methods, such as mirroring, rotation, brightness adjustment, and contrast adjustment. As shown in Table 3, the numbers of each category were 5342 images of Alternaria_Boltch, 5655 images of Brown_Spot, 4810 images of Gray_Spot, 4875 images of Mosaic, and 5694 images of Rust. Samples of the PlantVillage dataset and the Apple Leaf Pathology dataset are shown in Figure 7 and Figure 8, respectively. Some of the augmented images in the Apple Leaf Pathology dataset are shown in Figure 9.

Among all the crop disease datasets, the PlantVillage dataset has been selected by most research works to assess the multiclassification recognition capability of the model because of its large data size and multiple classifications. In the experiments, our proposed model was compared with other research works on the PlantVillage dataset. In addition, the ability to generalize on small datasets is an important basis for the robustness of the model. The Apple Leaf Pathology dataset with an original size of 1591 images was expanded more than 10 times after data augmentation, which is prone to model overfitting; i.e., the model achieves high accuracy on the training set but low accuracy on the testing set. Therefore, we choose the Apple Leaf Pathology dataset to demonstrate that the proposed model can also exhibit the high-accuracy generalization performance of small disease datasets.

## 3. Results and Analysis

### 3.1. Experiments on the PlantVillage Dataset

The models are trained on one NVIDIA RTX 3090 GPU with 24 GB RAM based on a 64-bit Ubuntu 20.04 operating system and PyTorch framework, with Python version 3.8.0, PyTorch version 1.10.0, and CUDA API version 11.3. The experimental parameters are set as follows: stochastic gradient descent (SGD) is used as the model optimizer, the initial learning rate is set to 0.01, the momentum is 0.9, the batch size of the model is set to 64, the number of iterations (epochs) is set to 200, and the loss function is CrossEntropy.

Several evaluation metrics are used to evaluate the model, including accuracy, recall, precision, sensitivity, specificity, and F1-score. Accuracy is the proportion of correctly classified samples to the total number of samples, precision is the proportion of samples with positive predictions that are actually positive, and recall is the proportion of samples with positive predictions that are actually positive. Sensitivity (Sens) is the ratio of all positive cases to all positive cases identified, and specificity (Spec) is the ratio of negative cases to all negative cases identified. F1-score takes recall and precision into account and is a weighted average of the two. The calculation formula for each metric is as Equations (5)–(10):(5)$$\mathrm{Accuracy}=\frac{\mathrm{TP}+\mathrm{TN}}{\mathrm{TP}+\mathrm{TN}+\mathrm{FP}+\mathrm{FN}}$$
(6)$$\mathrm{Recall}=\frac{\mathrm{TP}}{\mathrm{TP}+\mathrm{FN}}$$
(7)$$\mathrm{Precision}=\frac{\mathrm{TP}}{\mathrm{TP}+\mathrm{FP}}$$
(8)$$\mathrm{Sensitivity}=\frac{\mathrm{TP}}{\mathrm{TP}+\mathrm{FN}}$$
(9)$$\mathrm{Specificity}=\frac{\mathrm{TN}}{\mathrm{FP}+\mathrm{TN}}$$
(10)$$F1-\mathrm{score}=\frac{2\mathrm{TP}}{2\mathrm{TP}+\mathrm{FP}+\mathrm{FN}}$$
where TP is the number of true-positive samples, FP is the number of false-positive samples, FN is the number of false-negative samples, and TN is the number of true-negative samples. In order to examine the lightness of the model, params and FLOPs are used as metrics in the experiments. Params denote the number of model parameters. FLOPs denote floating point operations number, understood as the amount of computation. FLOPs are currently used in most studies to measure the computational speed of an algorithm/model. When the FLOPs are smaller, it means that the model has a smaller computation, and therefore, the model has a higher computing speed.

We compared the performance of our method with the ResNet family of the ResNet18, ResNet34, ResNet50, and ResNet101 networks on the PlantVillage dataset. As shown in Figure 10, MSCVT shows a faster convergence rate than the ResNet network at the beginning of training, and the loss is kept lower during the training process, which indicates that the model has a stronger data fitting ability. Figure 11 shows the accuracy variation curves of each model over 100 epochs on the testing dataset. As can be seen, MSCVT’s testing accuracy increased most rapidly at the beginning of the training process and outperformed the ResNet network in less than 10 epochs. After 20 epochs, the model’s testing accuracy plateaued and was slightly ahead of the rest of the models. These results show that MSCVT has a better convergence speed than the ResNet network family. Table 4 shows the metrics of MSCVT on the training and testing sets, and it can be seen that the performance of the model on the training set is lower than that on the testing set. This indicates that the proposed model does not suffer from overfitting and exhibits excellent generalization ability. As can be seen in Table 5 and Table 6, after 200 epochs of training, the proposed MSCVT model achieved better performance than the ResNet family of networks with a minimum number of parameter values. The accuracy of the ResNet models differed for different depths, with ResNet101 having the highest accuracy of 99.79%. ResNet18, ResNet34, and ResNet50 are followed by accuracies of 99.71%, 99.75%, and 99.70%, respectively. On the one hand, the accuracy of MSCVT is found to be 0.15%, 0.11%, 0.16%, and 0.07% higher than the accuracy of ResNet18, ResNet34 ResNet50, and ResNet101, respectively. On the other hand, Table 6 shows that the number of parameters of MSCVT is roughly $\frac{1}{3},\frac{1}{5},\frac{1}{6}$, and $\frac{1}{10}$ of those of ResNet18, ResNet34, ResNet50, and ResNet101, respectively. In terms of speed, MSCVT also obtained lower FLOPs than the ResNet model, indicating that the model has less computational effort as well as higher running speed. In addition, the proposed model achieves the best performance in other evaluation metrics, such as sensitivity, specificity, recall, and F1-score. Thus, MSCVT obtains high accuracy performance without causing an excessive number of parameters.

Next, MSCVT was compared with the traditional heavyweight models VGG16 and VGG19 and the lightweight models MobileNetV1 and MobileNetV2. As shown in Figure 12 and Figure 13, our model shows a comparable convergence speed with the other models during the training of the first 100 epochs. Table 7 shows the final evaluation metrics for all models, and the proposed model reaches the most optimal. While the accuracies of VGG16 and VGG19 are similar to that of our model, their parameters and FLOPs are by far several tens of times higher. As shown in Table 8, MSCVT achieves a larger performance improvement at the cost of a small increase in parameters and FLOPs compared with the lightweight MobileNetV1 and MobileNetV2 networks. In addition, MSCVT has the best recall, sensitivity, specificity, and F1-score, indicating that it is more powerful in identifying positive samples and more difficult to misclassify, which reflects that the proposed model is more powerful in terms of comprehensive performance.

We also performed a detailed comparison of MSCVT with models used by other researchers. Among the models used for comparison are classical CNN models, such as VGG and GoogleNet, the lightweight MobileNetV1 and MobileNetV2 models, and some improved versions. For fairness, models used for comparison need to be trained and tested on PlantVillage’s dataset. As shown in Table 9, our proposed method provides the best accuracy in comparison with different types of CNN models. Therefore, the high performance of MSCVT is convincing. In addition, to demonstrate the capability of MSCVT to identify multiple crop diseases, we extracted nine types of diseased leaves and six types of healthy leaves from the testing dataset to draw the confusion matrix. The confusion matrix can represent the number of correct identifications and the number of misclassifications of the model in each category. Figure 14 shows that our proposed model achieves high accuracy performance on all classifications, showing the robustness of MSCVT on the task of classification of multiple diseases.

### 3.2. Experiments on the Apple Leaf Pathology Dataset

In the same way, on the smaller Apple Leaf Pathology dataset, we still compared the ResNet family of networks with the proposed MSCVT experimentally. As shown in Figure 15 and Figure 16, all models converge within 100 epochs. In the early stage of training, our model accuracy rose significantly faster than the other models and maintained the highest accuracy rate. Table 10 shows that the proposed model also does not show overfitting on smaller-sized datasets, showing a strong generalization ability. The various evaluation metrics are summarized in Table 11. The results show that MSCVT performs best with 97.50% accuracy and achieves the best performance on recall, precision, sensitivity, specificity, and F1-score with 97.52%, 97.51%, 97.52%, 99.36%, and 97.51%, respectively. This indicates that our model still maintains higher positive sample recognition and a lower false-positive rate on a smaller-sized dataset. Among the ResNet family of networks, ResNet18 and ResNet34 both achieved 97.19% accuracy, outperforming ResNet50 and ResNet101 with deeper structures. Compared with ResNet18, ResNet34, ResNet50, and ResNet101, the accuracy of MSCVT is improved by 0.31%, 0.31%, 0.65%, and 0.48%.

As shown in Figure 17 and Figure 18, our model achieves a comparable convergence rate to other CNN models, including the light and heavy models on the Apple Leaf Pathology dataset. Table 12 also demonstrates that the proposed model achieves advanced recognition performance in all evaluation metrics. However, MSCVT obtains a recognition performance close to that of VGG19 with a number of parameters tens of times lower. In comparison with the heavyweight model VGG16, the accuracy of MSCVT is improved by 0.25%. On the other hand, MSCVT improves accuracy by 1.90% and 0.23% compared with the lightweight models MobileNetV1 and MobileNetV2, respectively. In recall, precision, sensitivity, specificity, and F1-score evaluation metrics, our model obtains a performance second only to VGG19. In terms of the model size and speed, according to Table 13 and Table 14, the proposed model is second only to MobileNetV1 and MobileNetV2.

In addition, we plotted the confusion matrix based on the whole testing dataset. Figure 19 shows that MSCVT kept the number of false positives within 50 for all 5 classifications on the Apple Leaf Pathology dataset. After the statistical operation, the recognition rates of Alternaria_Boltch, Brown_Spot, Grey_Spot, Mosaic, and Rust on the five apple leaf diseases were 96.79%, 98.76%, 96.43%, 100%, and 96.38%, respectively. These results show that the model maintains high accuracy in the identification of each apple disease without severe recognition bias. In summary, one can conclude that MSCVT can still achieve state-of-the-art recognition performance on small-scale pathological datasets.

### 3.3. Ablation Experiments for SA Module

Class activation mapping (CAM) is a technique for visualizing the image features extracted by a model, which can show how much attention the model pays to each region of the image [41]. A darker red region means that the model is more interested in that region and considers it more likely to be a target for recognition. Conversely, if a region is darker in blue, it means that the region tends to be neglected. We selected MSCVT, ResNet101, VGG19, and MobileNetV2 for visualization and comparison. Figure 20 and Figure 21 show the visualization results of different models on diseased leaves. The proposed model can be found to capture the most complete diseased regions of the leaf, whereas other models are more likely to ignore disease characteristics. In addition, our model is able to focus well on the key disease information and filter out the nonfocused information. In this way, the initial signal can be obtained that the SA module in MSCVT broadens the receptive field of the model and allows the model to focus on more disease features. Next, the SA module is used for the ablation experiments.

To highlight the ability of the SA module in capturing global information, we use CAM to visualize the effects produced by the SA module. After removing the SA module from MSCVT, the differences of the feature maps are analyzed. Figure 22 shows that the original model pays more attention to the edge areas of the image than the model with the SA module removed. When the onset part of the leaf is located at the edge of the image, the SA module can significantly improve the sensitivity of the model to capture disease information. This phenomenon is explained by the fact that the self-attention mechanism provides a global receptive field, which allows the model to acquire feature information at a distance in the image.

Previous CAM visualization experiments have demonstrated the effectiveness of the self-attention mechanism in the process of crop disease identification by the model. To verify the effect of the SA module on the model performance, we compared the recognition accuracy for this module. The experiments were conducted on the PlantVillage dataset and the Apple Leaf Pathology dataset, respectively. The performance of both the original model and the model with the SA module removed on the two different datasets is shown in Table 15. When the SA module was added, the accuracy of MSCVT improved by 0.06% and 0.08% on the PlantVillage dataset and the Apple Leaf Pathology dataset, respectively. This result shows that with the stacking of multiscale CNN layers, our model already has a good recognition capability for most of the disease images. Even without adding the self-attention mechanism, the model still achieves comparable performance with other CNN models. However, there are a small proportion of images with insignificant edge features that are difficult to discriminate. The SA module can expand the receptive range of the model while maintaining the efficient inductive bias capability of the CNN layer, thus recognizing more disease features located at the edges of the images.

## 4. Conclusions

In this paper, by combining the benefits of CNN and ViT, we propose a new network MSCVT based on the ResNet architecture, which has a low number of parameters and advanced recognition accuracy for effective crop disease identification. The network is based on the five-layer stage structure of the ResNet network with SGD as the optimizer, a learning rate of 0.01, and a batch size of 64. After 200 epochs of training, the network achieves high recognition accuracy with fewer parameters on two crop datasets. On the work of model compression, the lightweight module of MobileNetV2, an inverted residual block, has been incorporated into our model, which can significantly reduce the cost of parameters required in convolutional operations. For recognition accuracy improvement, we propose the MSSA module, which uses multiscale convolution layers to identify localized disease features and the SA module to enhance the ability of the model to capture disease information at edge locations. Experimental results show that MSCVT exhibits significant recognition effects on the PlantVillage dataset and can achieve accurate recognition of multiple crop diseases. It achieves state-of-the-art performance compared with different types of CNN models. In addition, the proposed model achieved advanced recognition accuracy on the small-scale Apple Leaf Pathology dataset. The proposed model meets the demand for high-precision agricultural disease identification, can accurately identify a variety of crop diseases, makes up for the shortcomings of modern deep learning models for plant disease analysis and processing, effectively reduces human identification errors, and saves the time cost of technicians and experts arriving for identification because agricultural areas are far from cities. Additionally, its lightweight feature also meets the demand for mobile intelligent recognition in actual agricultural production, can quickly identify leaf disease images, and significantly improves the efficiency of pest and disease identification. Future research will attempt to deploy the model on mobile devices for real-time crop disease identification. Additionally, we hope to apply the model to other crop identification tasks.

## Figures and Tables

**Figure 1 sensors-23-06015-f001:**
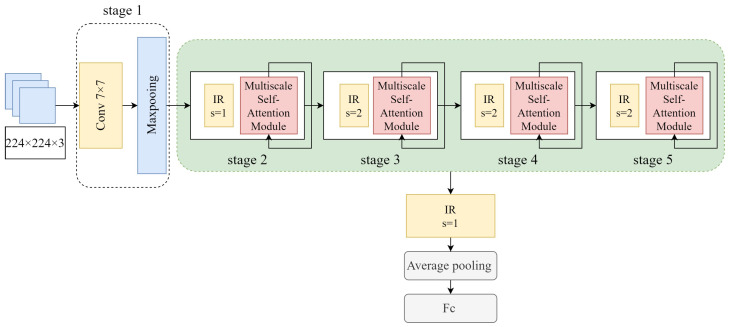
Model structure. The stride of the convolution operation is denoted as s.

**Figure 2 sensors-23-06015-f002:**
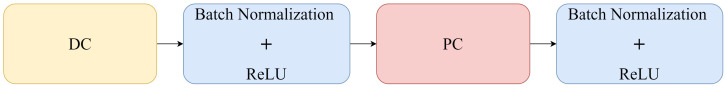
Structure of DSC.

**Figure 3 sensors-23-06015-f003:**
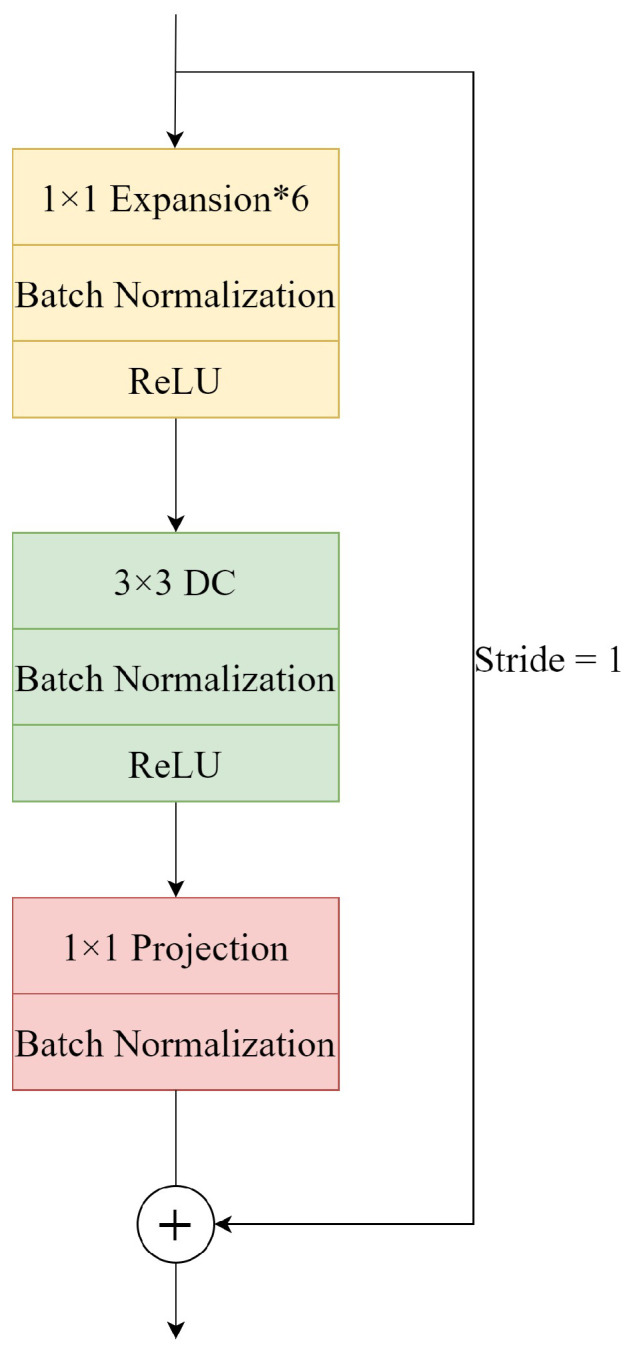
Structure of an inverted residual block. Expansion*6 denotes a channel expansion operation with the multiplier set to 6. Projection denotes the channel compression operation. Residual concatenation is only required when stride is 1. There is no residual concatenation for the downsampling layers in each stage.

**Figure 4 sensors-23-06015-f004:**
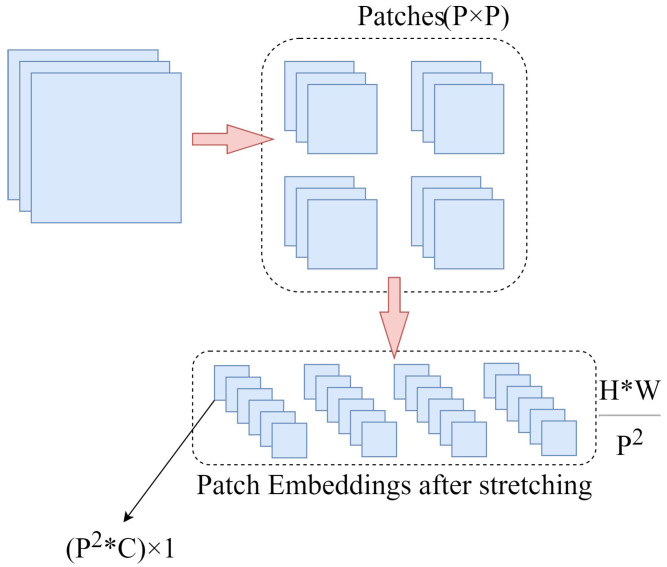
Patch Embedding. H, W, and C denote height, width, and the number of channels, respectively. * denotes multiplication between numbers.

**Figure 5 sensors-23-06015-f005:**
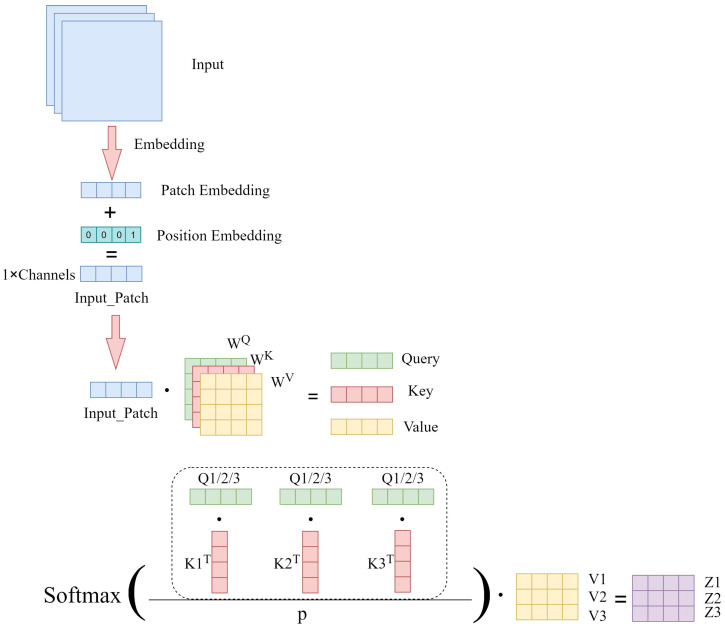
Self-attention module.

**Figure 6 sensors-23-06015-f006:**
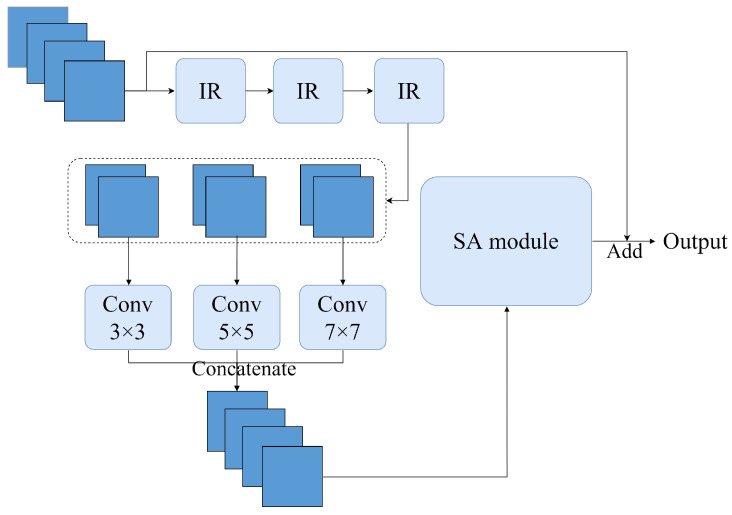
MSSA module.

**Figure 7 sensors-23-06015-f007:**
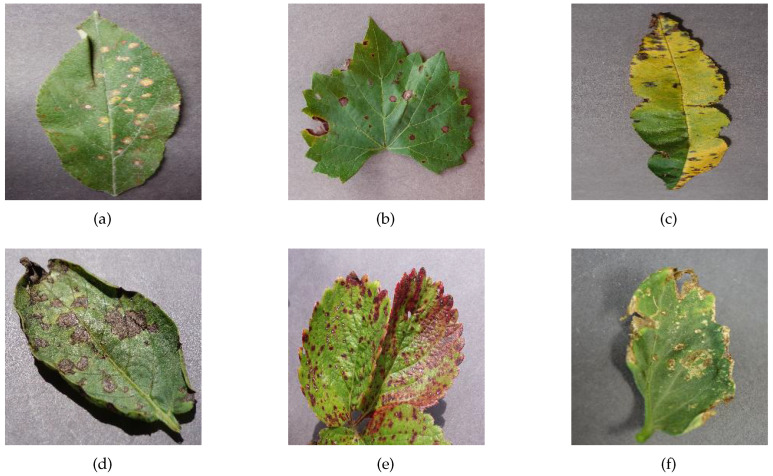
Examples of diseases in the PlantVillage dataset: (**a**) Apple_Cedar_apple_rust, (**b**) Grape_Black_rot, (**c**) Peach_Bacterial_spot, (**d**) Potato_Early_blight, (**e**) Strawberry_Leaf_scorch, (**f**) Tomato_Bacterial_spot.

**Figure 8 sensors-23-06015-f008:**
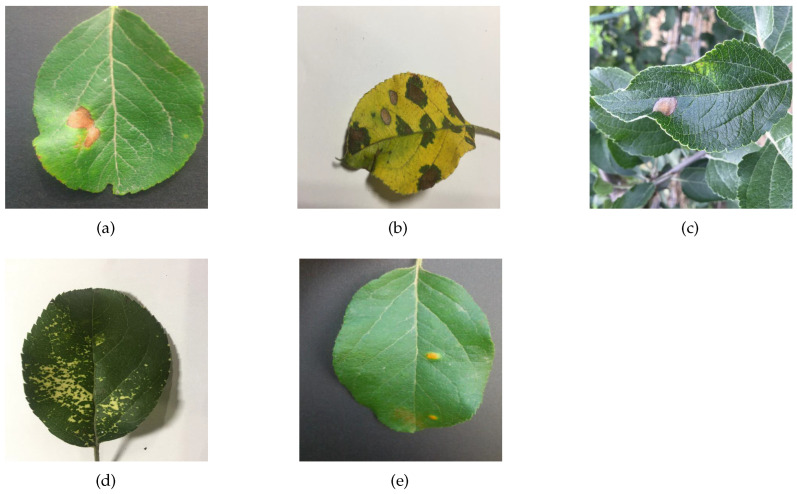
Some samples of Apple Leaf Pathology dataset: (**a**) Alternaria_Boltch, (**b**) Brown_Spot, (**c**) Grey_Spot, (**d**) Mosaic, (**e**) Rust.

**Figure 9 sensors-23-06015-f009:**
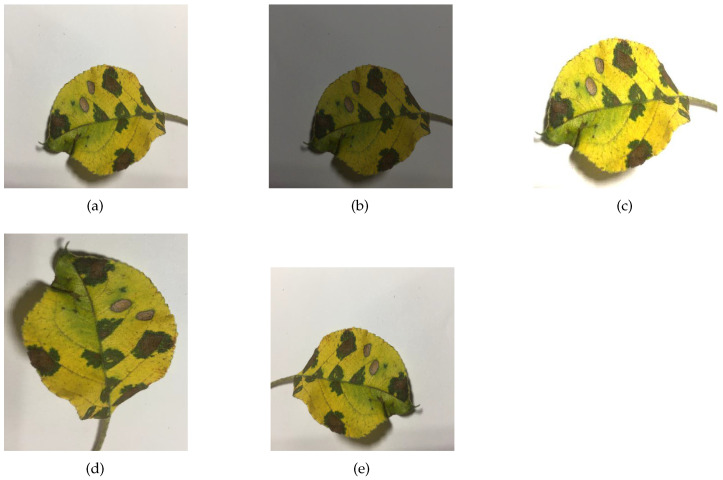
Some of the augmented images: (**a**) origin image, (**b**) turn down brightness, (**c**) enhance brightness, (**d**) rotation $90^{\circ }$, (**e**) horizontal flip.

**Figure 10 sensors-23-06015-f010:**
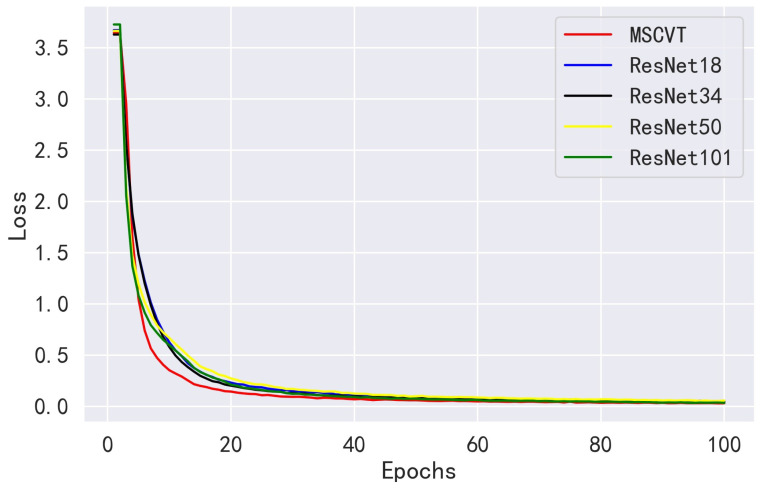
Loss variation curves of ResNet models on the PlantVillage training dataset.

**Figure 11 sensors-23-06015-f011:**
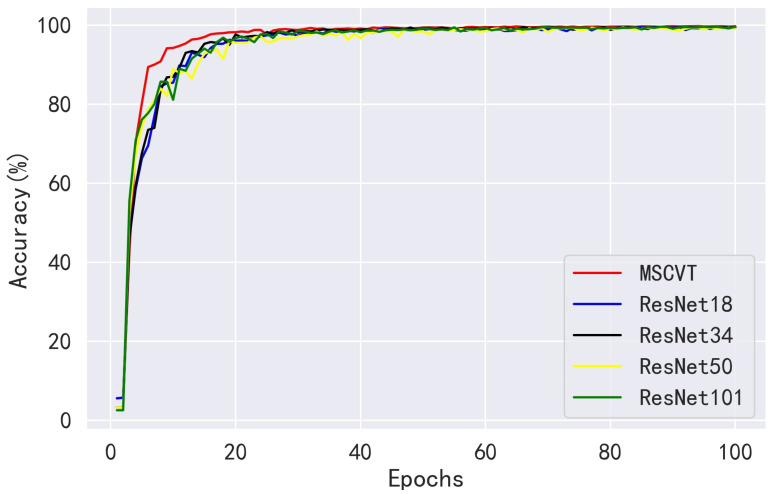
Accuracy variation curves of ResNet models on the PlantVillage testing dataset.

**Figure 12 sensors-23-06015-f012:**
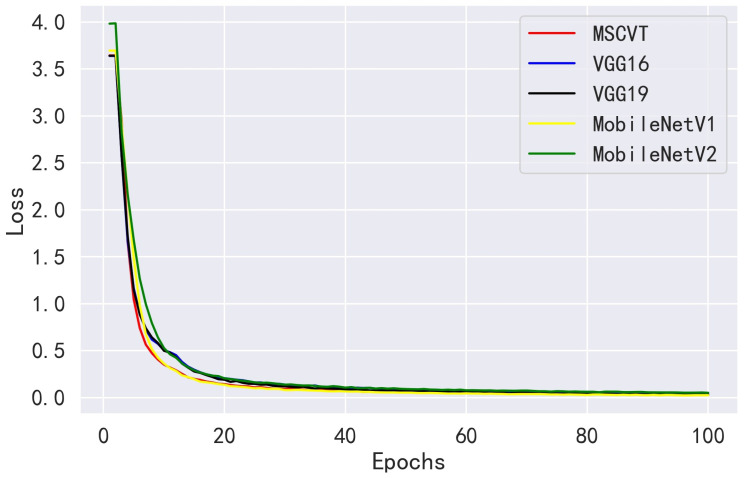
Loss variation curves of other CNN models on the PlantVillage training dataset.

**Figure 13 sensors-23-06015-f013:**
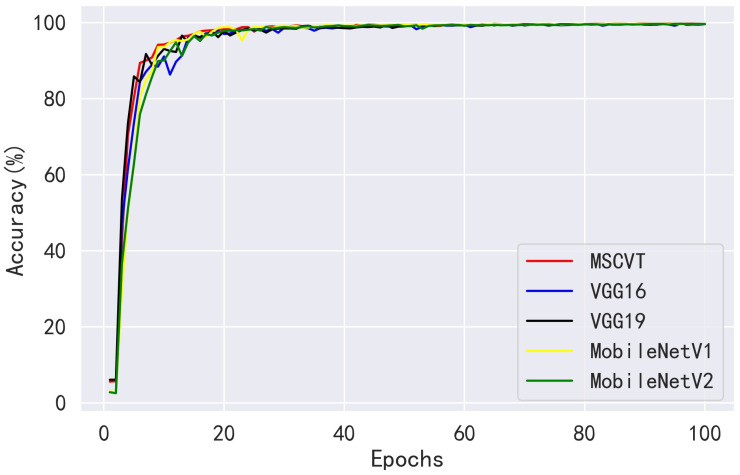
Accuracy variation curves of other CNN models on the PlantVillage testing dataset.

**Figure 14 sensors-23-06015-f014:**
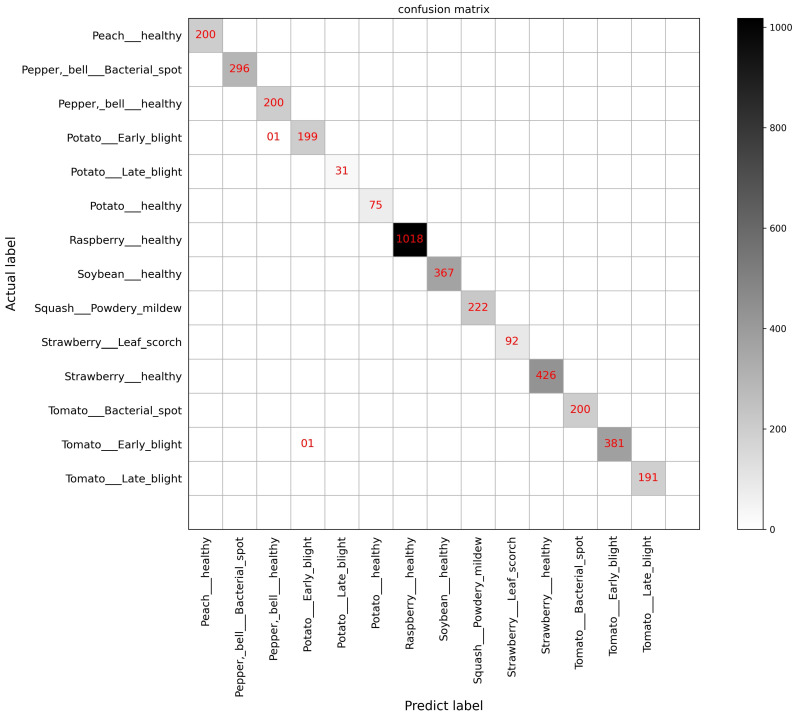
Confusion matrix on the PlantVillage dataset.

**Figure 15 sensors-23-06015-f015:**
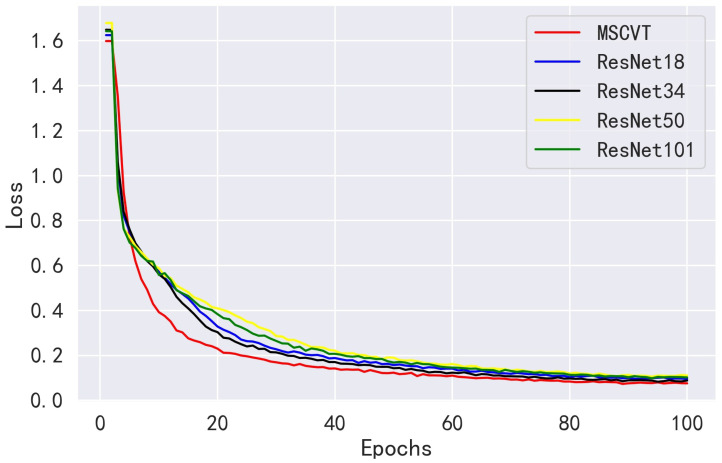
Loss variation curves of ResNet models on the Apple Leaf Pathology training dataset.

**Figure 16 sensors-23-06015-f016:**
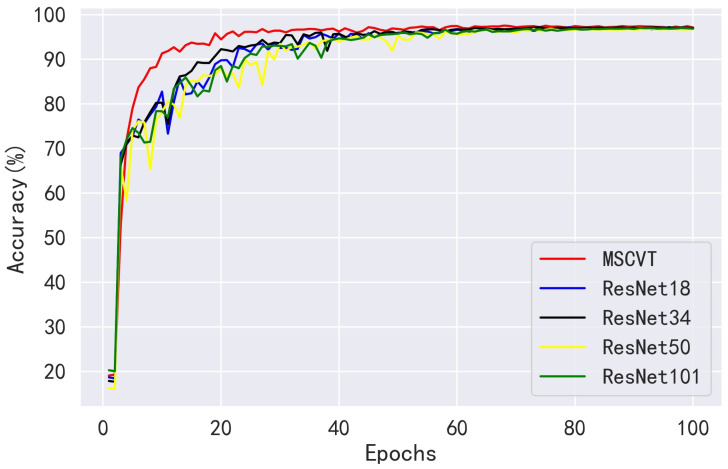
Accuracy variation curves of ResNet models on the Apple Leaf Pathology testing dataset.

**Figure 17 sensors-23-06015-f017:**
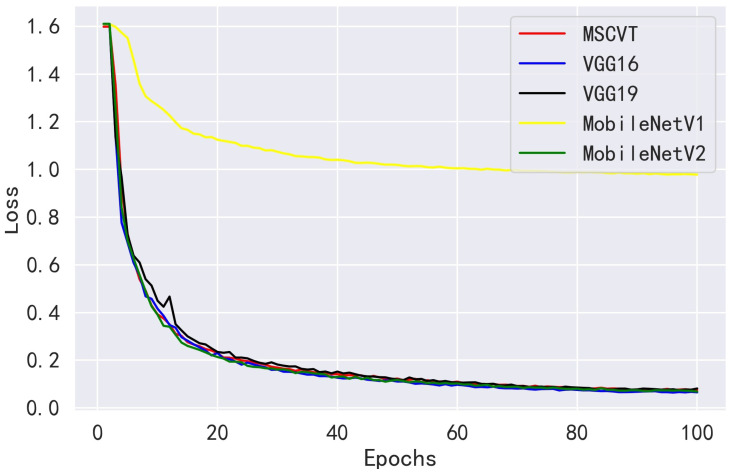
Loss variation curves of other CNN models on the Apple Leaf Pathology training dataset.

**Figure 18 sensors-23-06015-f018:**
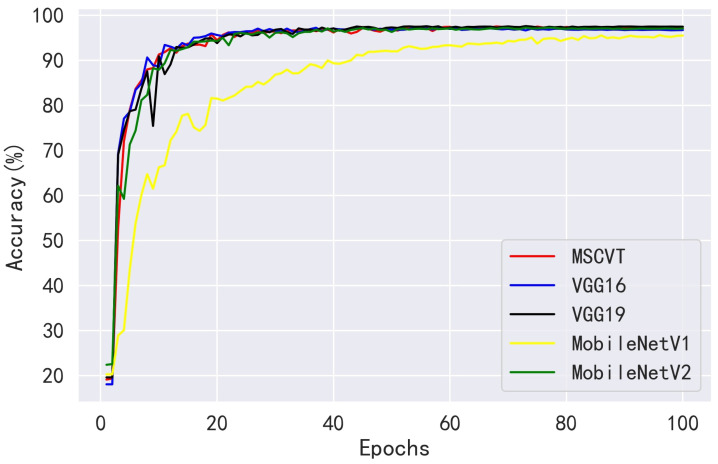
Accuracy variation curves of other CNN models on the Apple Leaf Pathology testing dataset.

**Figure 19 sensors-23-06015-f019:**
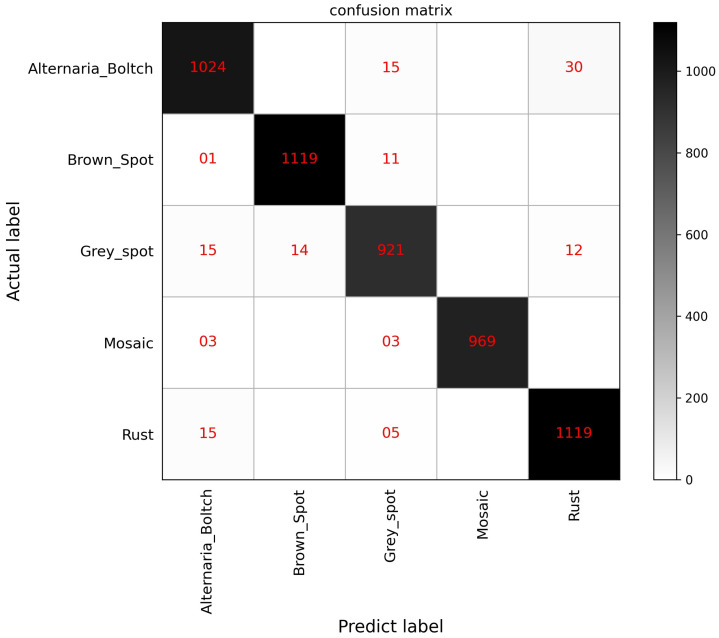
Confusion matrix on the Apple Leaf Pathology testing dataset.

**Figure 20 sensors-23-06015-f020:**
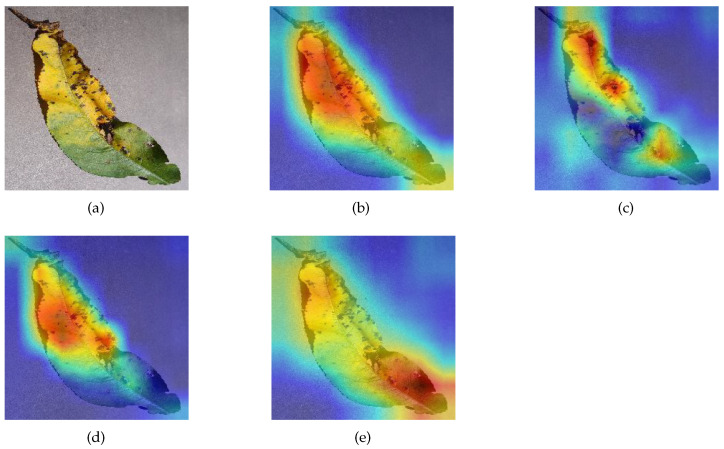
Visualization. Darker red means the model pays more attention to the region, and darker blue means the model ignores the region more. results on Peach_Bacterial_spot: (**a**) origin image, (**b**) MSCVT, (**c**) VGG19, (**d**) ResNet101, (**e**) MobileNetV2.

**Figure 21 sensors-23-06015-f021:**
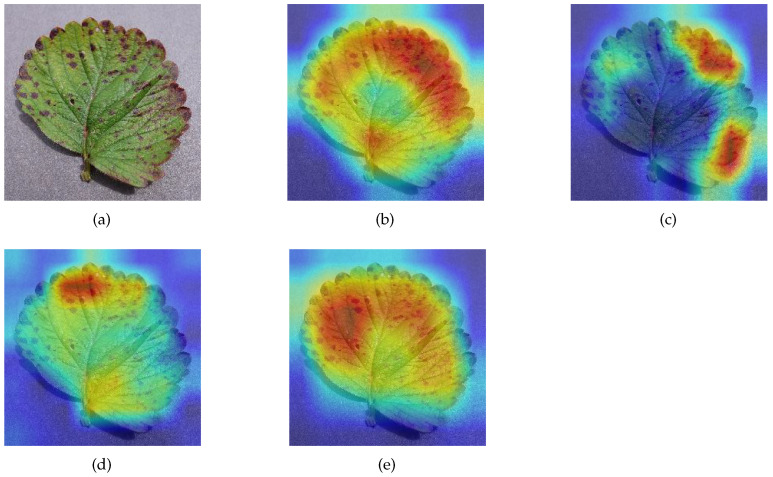
Visualization. Darker red means the model pays more attention to the region, and darker blue means the model ignores the region more. results on Strawberry_Leaf_scorch, (**a**) origin image, (**b**) MSCVT, (**c**) VGG19, (**d**) ResNet101, (**e**) MobileNetV2.

**Figure 22 sensors-23-06015-f022:**
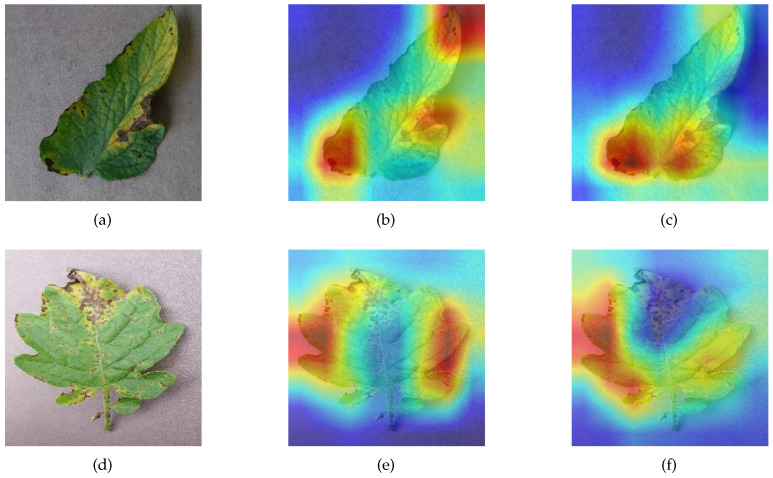
Visualization. Darker red means the model pays more attention to the region, and darker blue means the model ignores the region more. results of SA module: (**a**) sample1 (Tomato_Early_blight), (**b**) MSCVT for sample1, (**c**) MSCVT without SA module for sample1, (**d**) sample2 (Tomato_Septoria_leaf_spot), (**e**) MSCVT for sample2, (**f**) MSCVT without SA module for sample2.

**Table 1 sensors-23-06015-t001:** Input shapes and output shapes (height× width × channel) of each stage. k is denoted the number of disease categories.

	Input Shapes	Output Shapes	Strides
Stage 1	224 × 224 ×3	56 × 56 × 24	2
Stage 2	56 × 56 × 24	56 × 56 × 32	1
Stage 3	56 × 56 × 32	28 × 28 × 48	2
Stage 4	28 × 28 × 48	14 × 14 × 88	2
Stage 5	14 × 14 × 88	7 × 7 × 168	2
IR	7 × 7 × 168	7 × 7 × 320	1
Average pooling	7 × 7 × 320	1 × 1 × 320	-
Fc	1 × 1 × 320	k × 1	-

**Table 2 sensors-23-06015-t002:** Distribution of the PlantVillage dataset.

Crop Category	Number of Disease Types	Number of Health Categories	Number of Images
Tomato	9	1	18,160
Orange	1	0	5507
Soybean	0	1	5090
Grape	3	1	4063
Corn	3	1	3852
Apple	3	1	3171
Peach	1	1	2657
Pepper	1	1	2475
Potato	2	1	2152
Cherry	1	1	1906
Squash	1	0	1835
Strawberry	1	1	1565
Blueberry	0	1	1502
Raspberry	0	1	371
Total	26	12	54,306

**Table 3 sensors-23-06015-t003:** Distribution of the Apple Leaf Pathology dataset.

Crop Category	Number of Images
Rust	5694
Brown_Spot	5655
Alternaria_Boltch	5342
Mosaic	4875
Gray_Spot	4810
Total	26,376

**Table 4 sensors-23-06015-t004:** Comparison of testing performance and training performance of the MSCVT on PlantVillage dataset.

Dataset	Accuracy (%)	Recall (%)	Precision (%)	F1-Score (%)	Sens (%)	Spec (%)
Train	99.59	99.35	99.43	99.46	99.43	99.98
Test	99.86	99.82	99.72	99.77	99.82	99.99

**Table 5 sensors-23-06015-t005:** The performance of ResNet models on the PlantVillage testing dataset.

Model	Accuracy (%)	Recall (%)	Precision (%)	F1-Score (%)	Sens (%)	Spec (%)
MSCVT	99.86	99.82	99.72	99.77	99.82	99.99
ResNet18	99.71	99.66	99.52	99.55	99.66	99.98
ResNet34	99.75	99.72	99.62	99.63	99.72	99.98
ResNet50	99.70	99.58	99.58	99.58	99.58	99.98
ResNet101	99.79	99.65	99.74	99.69	99.65	99.99

**Table 6 sensors-23-06015-t006:** The lightweight indicators of ResNet models on the PlantVillage testing dataset.

Model	Param (M)	FLOPs (M)
MSCVT	4.20	1035.78
ResNet18	11.20	1823.52
ResNet34	21.30	3678.22
ResNet50	23.59	4131.69
ResNet101	42.57	7864.39

**Table 7 sensors-23-06015-t007:** The performance of other CNN models on the PlantVillage testing dataset.

Model	Accuracy (%)	Recall (%)	Precision (%)	F1-Score (%)	Sens (%)	Spec (%)
MSCVT	99.86	99.82	99.72	99.77	99.82	99.99
VGG16	99.83	99.78	99.76	99.77	99.78	99.99
VGG19	99.84	99.75	99.77	99.76	99.75	99.99
MobileNetV1	99.71	99.60	99.61	99.60	99.60	99.98
MobileNetV2	99.82	99.79	99.66	99.72	99.79	99.99

**Table 8 sensors-23-06015-t008:** The lightweight indicators of other CNN models on the PlantVillage testing dataset.

Model	Param (M)	FLOPs (M)
MSCVT	4.20	1035.78
VGG16	134.41	15,466.18
VGG19	139.72	19,627.97
MobileNetV1	3.26	587.93
MobileNetV2	2.44	542.18

**Table 9 sensors-23-06015-t009:** The performance of different studies on the PlantVillage dataset.

Study	Year	Model (%)	Accuracy (%)
Mohanty et al. [14]	2016	GoogleNet	99.34
Ferentinos [38]	2018	VGG	99.53
Kamal et al. [20]	2019	MobileNet	98.65
Kamal et al. [20]	2019	Reduced MobileNet	98.34
Gao et al. [39]	2021	DECA_ResNet18	99.74
Sanida et al. [40]	2021	MobileNetV2	98.08
Sutaji and Yıldız [22]	2022	LEMOXINET	99.10
This study	2023	MSCVT	99.86

**Table 10 sensors-23-06015-t010:** Comparison of testing performance and training performance of MSCVT on the Apple Leaf Pathology dataset.

Dataset	Accuracy (%)	Recall (%)	Precision (%)	F1-Score (%)	Sens (%)	Spec (%)
Train	96.74	96.74	96.74	96.86	96.74	99.18
Test	97.50	97.52	97.51	97.51	97.52	99.36

**Table 11 sensors-23-06015-t011:** The performance of ResNet models on the Apple Leaf Pathology testing dataset.

Model	Accuracy (%)	Recall (%)	Precision (%)	F1-Score (%)	Sens (%)	Spec (%)
MSCVT	97.50	97.52	97.51	97.51	97.52	99.36
ResNet18	97.19	97.19	97.19	97.19	97.19	99.26
ResNet34	97.19	97.25	97.19	97.21	97.25	99.28
ResNet50	96.85	96.89	96.84	96.86	96.89	99.21
ResNet101	97.02	97.06	97.01	97.04	97.01	99.25

**Table 12 sensors-23-06015-t012:** The performance of other CNN models on the Apple Leaf Pathology testing dataset.

Model	Accuracy (%)	Recall (%)	Precision (%)	F1-Score (%)	Sens (%)	Spec (%)
MSCVT	97.50	97.52	97.51	97.51	97.52	99.36
VGG16	97.25	97.26	97.27	97.26	97.26	99.21
VGG19	97.63	97.62	97.66	97.64	97.62	99.38
MobileNetV1	95.60	95.56	95.53	95.53	95.26	99.02
MobileNetV2	97.27	97.32	97.28	97.30	97.32	99.27

**Table 13 sensors-23-06015-t013:** The lightweight indicators of ResNet models on the Apple Leaf Pathology testing dataset.

Model	Param (M)	FLOPs (M)
MSCVT	4.18	1035.76
ResNet18	11.18	1823.52
ResNet34	21.29	3678.22
ResNet50	23.52	4131.69
ResNet101	42.51	7864.38

**Table 14 sensors-23-06015-t014:** The lightweight indicators of other CNN models on the Apple Leaf Pathology testing dataset.

Model	Param (M)	FLOPs (M)
MSCVT	4.20	1035.76
VGG16	134.28	15,466.17
VGG19	139.59	19,627.97
MobileNetV1	3.22	587.89
MobileNetV2	2.40	542.13

**Table 15 sensors-23-06015-t015:** Ablation experiment results for SA module.

Model	Plant Village Dataset (%)	Apple Leaf Pathology Dataset (%)
MSCVT	99.86	97.50
MSCVT without SA module	99.80	97.42

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
