# Peer review of "Crop Disease Identification by Fusing Multiscale Convolution and Vision Transformer"

_sensors, 2023, doi:10.3390/s23136015_

Round 1

Reviewer 1 Report

The research aimed to develop a deep learning model for disease-recognition. The topic is original and relevant to the field and it addresses a specific gap in the field (developing a deep learning model for disease-recognition in agriculture). I would suggest the authors provide more information on the model evaluation. For example, train/test data model assessment (Accuracy, No information rate, sensitivity and specificity percentages). The references could be improved.

I would suggest to provide information on the general machine learning algorithms and discuss their advantages and disadvantages, and whit which data types they are appropriate to be used. For this information you can refer and cite the following literature: 

Akin, M., Eyduran, S. P., Eyduran, E., & Reed, B. M. (2020). Analysis of macro nutrient related growth responses using multivariate adaptive regression splines. Plant Cell, Tissue and Organ Culture (PCTOC)140, 661-670.

Reviewer 2 Report

The article presents a method for identifying leaf diseases using the ResNet architecture. The article is interesting and relevant. To improve the quality of the article, the following changes should be made:

1.) The conclusions should describe the hyperparameters of the model in more detail.

2.) The practical significance of the obtained research results should be described in more detail.

3.) The speed characteristics (FPS) of the compared models should be evaluated in more detail in Section3.

4.) In section 1 of the article, an analysis of the use of convolutional neural networks, for example, the well-known YOLO models for solving the problem of identifying leaf diseases should be added.

5.) Which service is used for augmentation of the data set (Fig.9)?

6.) It is necessary to describe in more detail the quantitative and qualitative characteristics of the data set used.
